# Epidemiological Analysis of Fungal Infection Disease in Pediatric Population: Focus on Hospitalization from 2007 to 2022 in Veneto Region in Italy

**DOI:** 10.3390/pathogens14010093

**Published:** 2025-01-18

**Authors:** Lorenzo Chiusaroli, Claudia Cozzolino, Silvia Cocchio, Mario Saia, Carlo Giaquinto, Daniele Donà, Vincenzo Baldo

**Affiliations:** 1Division of Pediatric Infectious Diseases, Department for Women’s and Children’s Health, University of Padua, 35126 Padua, Italy; carlo.giaquinto@unipd.it (C.G.); daniele.dona@unipd.it (D.D.); 2Department of Cardiac, Thoracic, Vascular Sciences and Public Health, University of Padua, 35126 Padua, Italy; claudia.cozzolino@studenti.unipd.it (C.C.); silvia.cocchio@unipd.it (S.C.); vincenzo.baldo@unipd.it (V.B.); 3Preventive Medicine and Risk Assessment Unit, Hospital-University of Padua, 35126 Padua, Italy; 4Azienda Zero of Veneto Region, 35131 Padua, Italy; mario.saia@azero.veneto.it

**Keywords:** fungal infection, fungal disease, epidemiology, children, hospitalization

## Abstract

Fungal infections (FIs) are widespread globally, affecting both immunocompromised and immunocompetent children, with varying clinical implications based on age and comorbidities. In immunocompromised children, particularly those with hematologic oncological conditions, FI leads to substantially longer hospital stays and increased in-hospital mortality, with reported rates ranging from 15% to 20%. Our study aims to analyze the epidemiological trends of fungal infections in the pediatric population within a specific region of Italy. We extracted ICD-9 codes related to fungal infections from hospital discharge records (HDRs) in the pediatric population of Veneto, located in the north-east of Italy, between 2007 and 2022. We included all children admitted to the hospital with a primary or secondary diagnosis during admission for other reasons. Data were stratified based on age, year, ward of admission, and type of diagnosis. Patients older than eighteen and HDRs related to a second admission within thirty days from the previous admission were excluded. A total of 1433 diagnoses were analyzed during the period, with 241 (16.8%) as main diagnoses and 1192 (83.2%) as secondary diagnoses. The overall hospitalization rate was 1084 cases/100,000 (1.69 cases/100,000 as primary diagnosis and 8.95 cases/100,000 as secondary). The hospitalization rate stratified for age was 11,055 cases/100,000 among infants younger than 1 year, 8.48 cases/100,000 among those aged 1-4 years, and 4.4 cases/100,000 among children older than 5. The more frequent infection was *Candida* spp. (62.8%), followed by *Aspergillus* spp. (14.6%) and skin mycosis (9.5%). Overall, the pediatric in-hospital case fatality rate due to FI was 2.09%. Our study elucidated the overall experience of fungal infections in the pediatric population of the Veneto region in Italy. Specifically, we underscored a relatively stable hospitalization rate for fungal diseases and a noteworthy mortality rate.

## 1. Background

Fungal infections (FIs) are common in all parts of the world, and in recent decades, many strategies have evolved, especially regarding the growing array of antifungal compounds. However, clinical situations could be differently related to the immune system of the patients: in immunocompromised children, especially with onco-hematological disease, invasive FIs are associated with significant increases in hospital lengths of stay and in-hospital mortality, with reported overall mortality between 15 and 20% for pediatric patients [1]. Among adult patients, the epidemiological trends are satisfactorily documented [2,3,4]. However, in the pediatric population, the epidemiology in the general population and population with comorbidities is still unclear [5,6,7,8].

Molds species, especially *Aspergillus* spp., are usually related to immunocompromised and cystic fibrosis patients. *Aspergillus* and other molds can be inhaled from the host’s environment and cause lung disease in the immunocompromised child and disseminate to other tissues. [9]. Regarding *Candida* spp. infections during pediatric age, the possible profiles can be represented by diaper dermatitis, dermatophytosis, oral candidiasis, candidemia, and invasive candidiasis, especially for immunocompromised patients [10,11]. For the last setting, the intensive care unit, post-surgical outcome, disruption of the gastrointestinal mucosa, and colonization of the central venous catheter are associated with a higher risk of *Candida* spp. infections [12,13]. Dissemination to secondary sites is reported in 10–20% of pediatric patients with candidemia, and severe sepsis or septic shock occurs in about 30%. For invasive candidemia, overall fatality rates range between 10% and 25% [12].

Our study aims to analyze the epidemiological trends of FIs in the pediatric population within a region of Italy.

## 2. Methods

### 2.1. Setting and Data Source

We conducted this retrospective study in the Veneto region, located in the north-east of Italy. In 2022, Veneto had an average population of 4.8 million, a mean age of 46.1 years, and 50.9% of its population were female [14]. In addition, the Italian National Health System is a publicly funded system primarily financed through general taxation and organized on a regional basis [15].

We analyzed the acute ordinary hospital discharge records (HDRs) of public hospitals from January 2007 to December 2022. HDRs include the following data: age, gender, type of the hospital, date of admission, length of stay from admission, comorbidities, and outcome. According to regional decree no. 118 dated December 23, 2016 [16], each HDR contained one primary diagnosis or first-listed diagnosis concerning the main condition identified during the patient’s hospital stay and up to five secondary diagnoses based on the diagnostic codes of the *International Classification of Diseases*, *Ninth Revision*, *Clinical Modification* (ICD-9-CM).

### 2.2. Population and Acute Admissions for Fungal Infection

We included all HDRs related to children aged 0 to 18 years old either admitted to the hospital with a primary diagnosis of fungal infection (FI) or with a secondary diagnosis of FI during admission for other reasons.

The volume of acute admissions for fungal infection diagnosis is analyzed in terms of the number of admissions and related dynamics recorded in the period of 2007–2022.

Hospital admissions for FI were identified by the ICD-9-CM diagnoses reported in the Appendix A. Considering the previous epidemiology, ICD-9-CM diagnoses related to endemic mycosis were not considered. Based on adjunctive ICD-9-CM diagnosis (see Appendix A), all cases were grouped into three main categories: (i) neonatal infection, (ii) immunocompromised children, or (iii) other children. The categories included all admissions related to ICD-9-CM associated with all diagnoses of fungal infection in children (ICD-9-CM).

Patients older than eighteen and HDRs related to a second admission within thirty days from the previous admission were excluded.

### 2.3. Hospitalization Rate and Length of Stay

Based on the total number of hospital admissions concerning Veneto child residents each year, annual hospitalization rates were calculated by dividing the annual number of hospitalizations by the size of the resident population (source Demo Istat) [17], and expressing the rates as hospitalizations per 100,000 population. Further, the population was stratified according to the diagnosis in prematurity, solid organ transplantation (SOT), onco-hematological disease, and cystic fibrosis; consequently, the incidence rate of FI was estimated for each comorbidity.

The length of hospital stays was calculated as the days elapsing between the dates of admission and discharge, and the mean hospital stay was calculated. The case fatality rate (CFR) was calculated by dividing the number of in-hospital deaths by the number of patients hospitalized with a diagnosis of FI, expressed as a percentage.

Secondary analysis was epidemiology of FI pre- and post-SARS-CoV-2 pandemic.

### 2.4. Estimated Costs

The estimated direct cost to the health care system for FI-associated hospital admission was calculated using the diagnosis-related groups (DRGs) of hospitalized patients. The DRGs depend on the patient’s ICD classification at the time of their discharge from the hospital, their age and gender, and the consumption of resources during their hospital stay. According to the DRG-based reimbursement system, every hospitalized patient belongs to a group of diagnostically homogeneous cases. Patients within each category are therefore similar in clinical terms and are expected to require the same level of hospital resources. Veneto’s expenses per DRG are defined in the regional tariff schedule. Costs were expressed in euros (€) [18]. As a result, patients in the same DRG are assigned the same reimbursement charges. All hospital stays were analyzed considering, for the same patients, only the first hospital admission. Any case of secondary hospitalization, transfer to other acute care institutions, and admission to rehabilitation institutions, associated with the same patient, was removed from the initial dataset for the years 2007 to 2022.

### 2.5. Statistical Analysis

Categorical variables were represented with frequencies and percentages, while continuous variables were summarized as means, standard deviations (std), medians, minimum–maximum values, and interquartile ranges (IQRs). Chi-square tests (or Fisher’s exact tests) and Student’s *t*-tests (or Wilcoxon signed-rank tests) were employed to assess differences in HDR variables between groups.

Significant trends over the period considered were assessed by conducting a Joinpoint regression, estimating the annual percent changes (APCs) and the average APC (AAPCs), a summary measure of the trend over a given fixed interval [19]. An AAPC of zero coincides with the hypothesis of a trend that is neither increasing nor decreasing.

The 95% confidence intervals (95% CIs) were calculated as appropriate. A *p*-value < 0.05 was considered significant. All data manipulations, analyses, and visualizations were performed using Python 3.8.18 and R 4.2.2.

## 3. Results

### 3.1. Hospitalization for FI

From 2007 to 2022 a total of 1433 admissions due to FI were recorded. The overall hospitalization rate was 10,839 per 100,000 in the pediatric population. Stratifying for age, 51.1 % were under one year of age (110,552 per 100,000), 29.3 % were between 1 and 9 years old (6625 per 100,000), and 19.6% were between 10 and 18 years old (4426 per 100,000) (Figure 1). The majority of episodes were presented during the center years of the study (2010–2014), with a peak in 2012 (13,600 per 100,000 cases) and a subsequent decreasing incidence rate of FI. The negative trend in hospitalization rate resulted significantly after 2011 (APC = −4.79, *p*-value = 0.0012) (Figure 2).

The inpatient rate by gender was 9536 admissions per 100,000 males and 11,053 admissions per 100,000 females.

Patients were also stratified by comorbidities showing an incidence rate that was different for each category. About 9.8% of FI-hospitalized children had a history of prematurity, 3.2% of solid organ transplantation, 18.8% of onco-hematological conditions, and 0.6% of cystic fibrosis.

The median annual rates of hospitalization stratified for comorbidities were 242,424/100,000 for patients with a history of prematurity; 140,3509/100,000 for people with a history of organ transplantation; 596,491/100,000 for patients with an onco-hematological history; and 26,315/100,000 for persons with a history of cystic fibrosis (Figure 3). For all patients with comorbidities, the hospitalization rate was significantly higher (*p*-value < 0.015) than those who did not present with comorbidities.

Regarding the length of hospitalization, the overall average was 14,7 days (median 6 days). Length of stay was significantly higher in 10–14-year-old patients (mean 27.8 days, median 11) and during the COVID-19 pandemic (18.2 versus 14.2, *p*-value = 0.0021). The trend analysis revealed an increasing pattern in the length of stay (AAPC = 6.17, *p*-value < 0.001) (Figure 4).

Stratifying for comorbidities, patients with a history of solid organ transplantation had the longest length of hospitalization (49.8 days), followed by patients with a history of cystic fibrosis (38.6 days) and patients with onco-hematological disease (26.1 days).

### 3.2. DRG Analysis

In the analyzed period, most of the diagnosis was represented by candidiasis (50.2%, ICD-9-CM from 11,281 to 11,289), neonatal *Candida* (12.1%, ICD-9-CM 7717), and aspergillosis infection (10.5%, ICD-9-CM 1173 and 4846). The complete list of diagnoses is reported in Table 1.

Candidiasis and neonatal *Candida* were prevalently represented among children younger than one year old (59.97% and 23.91, respectively), compared to aspergillosis infection among patients between 10 and 14 years old (28.93%).

The overall case fatality rate (CRF) was 2.094% (30 patients, *p*-value < 0.0001) lower for children < one year old (0.820), and higher for patients between 10 and 14 years old (4.403%). Mortality was significantly higher during the COVID-19 pandemic rather than in the previous years (3483 versus 1867). Overall, the Joinpoint regression found no trend (AAPC = 4.45, 95%CI −3.32; 1025); however, the 2019–2022 APC resulted in significant positive results (52.78, *p*-value = 0.042), highlighting a fast growth in CFR after 2020 (Figure 5).

Further, CFR was higher for patients with onco-hematological comorbidities (5926%) and post-solid organ transplantation (26,087%). In contrast, prematurity and cystic fibrosis were associated with lower CFR (0.709% and 0%, respectively).

### 3.3. Estimated Cost

The overall estimated hospitalization cost for FI was EUR 46,128 (median 27,768) with a mean estimated daily cost of 5659 euros/day (median 4082). The median cost was stable through the years (AAPC = 0.0018) with the highest value in 2014 (7489 euros/day), but without any peek during the COVID-19 pandemic. The average cost was 5504 euros/day for patients with a history of prematurity, 4791 euros/day for immunocompromised patients, 3951 euros/day for solid organ transplantation, and 3843 euros/day for cystic fibrosis. According to ICD-9-CM, the higher cost was associated with pneumonia due to non-aspergillosis (ICD-9-CM 4847 and 4848 with 8539 euros/day). Furthermore, surgical DRGs were associated with higher daily costs (8370 euros/day) (Figure 6).

All FI features and outcome assessments are reported in Table 2.

## 4. Discussion

Based on HDRs, the present retrospective study analyzed hospitalizations for FI in the Veneto region, Italy, for the period of 2007–2022 in the pediatric population. Admission for FI is mainly concentrated in the first year of age, followed by those who are 1–9 years old and 10–18 years old. *Candida* spp. is a documented infection in the neonatal phase of life with a variable incidence and mortality ranging from 12% to 37% in high-income countries (HICs) and from 8.9% to 75% in low- and middle-income countries (LMICs) [20], especially for prematurity with early gestational age and low weight [21]. However, in our study, we also analyzed changes in hospitalizations for FI from 2007 to 2022. Over this period, there was a slight reduction in hospitalizations for FI, with a progressive reduction from 13.6 in 2012 to 6.88 in 2022. A possible explanation for this reduction trend could be the use of antifungal prophylaxis in specific settings, such as neonatal intensive care units and onco-hematology departments where antifungal prophylaxis protocols are often applied. Compared to other studies in the literature, Marya et al. reported a hospitalization rate between 3,65 and 5.56/100,000 cases between 2000 and 2005 for *Candida* spp. infection [22] and a similar hospitalization rate is also analyzed in other studies [23]. For mold infections, the hospitalization rate is reported to be from 1.7 to 3.4 per million persons between 2000 and 2013 in adult studies [24]. Unfortunately, data are scarce in the literature for the pediatric population: Zaoutis et al. described the incidence of candidemia as 43 cases per 100,000 hospital admissions of children [1] and as 437/100,000 cases for invasive aspergillosis among immunocompromised pediatric admissions in the United States [25]. Both studies described a stable trend throughout the analyzed period.

In-hospital case fatality rate occurred in roughly 2.09% of cases. Considering age-related mortality, a higher case fatality rate has been observed in patients between 5 and 14 years old (4.4%) despite a relatively lower number of infections than neonatal age (0.82%). These data are considerably related to the higher number of oncological and solid organ transplantation patients. Stratifying for the pathogen, we noted higher CFR for *Aspergillus* spp. infections (14%) and relatively lower CFR for *Candida* spp. in particular for neonatal patients (0.6%). A similar high CFR for *Aspergillus* spp. is reported by the oncological pediatric population [25], indicating the seriousness of *Aspergillus* spp. infection in this population. However, in our analysis, an increasing trend of mortality was reported after the SARS-CoV-2 pandemic in 2020 (Figure 5) similar to the hospitalization length stay (Figure 6): Muthu suggested early-onset mold infections after COVID-19 and a more severe COVID-19 course in mold pneumonia co-infections in the adult population [26]. The potential causality of a prior viral infection remains unestablished. Some authors suggest a possible role for co-isolated infections in enhancing disease virulence, as well as an association between *Aspergillus* colonization and increased COVID-19 mortality [27,28]. Additionally, environmental factors could play a role, particularly in construction and renovation activities within hospitals or nearby areas. In Italy, and specifically in our region, a national building renovation plan has been underway since 2020, which might contribute to an increased number and severity of infections [29].

However, the real causality of these events is still unclear and further data are required.

In economic terms, the mean daily value of acute inpatient admission is EUR 565 with a maximum of EUR 13,063. The direct costs are not comparable with other studies for different values and situations [25].

Our study has several limitations. The first is the reliance on HDRs from the Veneto region, as these data are primarily used as an administrative tool and only secondarily used for healthcare purposes. Accessing complete medical histories was not feasible within the scope of this research. As a result, our classification depends on the diagnosis codes recorded in the HDRs, which are still coded using the ICD-9-CM system in the Veneto region. In addition to possible miscoding, we emphasize that our study may also be susceptible to underestimation of cases of fungal infection due to other etiologic agents or co-infection with other etiologic agents. Furthermore, HDR data do not contain any information about the microbiological, diagnostic, and therapeutic information, limiting the possible evaluation of the patient.

## 5. Conclusions

In summary, we reported a trend of FI hospitalizations in the Italian region from 2007 to 2022 in the pediatric population showing a general decrease despite persistently high case fatality rates, which increased following the COVID-19 pandemic. FI was more frequently observed in children under one year of age, primarily due to *Candida* species, while adolescents were more affected by *Aspergillus* and other mold infections. Significant differences are presented in hospitalizations for FI for comorbidities with higher impacts in terms of length of stay and CRF in oncological patients and those who had undergone solid organ transplantation.

## Figures and Tables

**Figure 1 pathogens-14-00093-f001:**
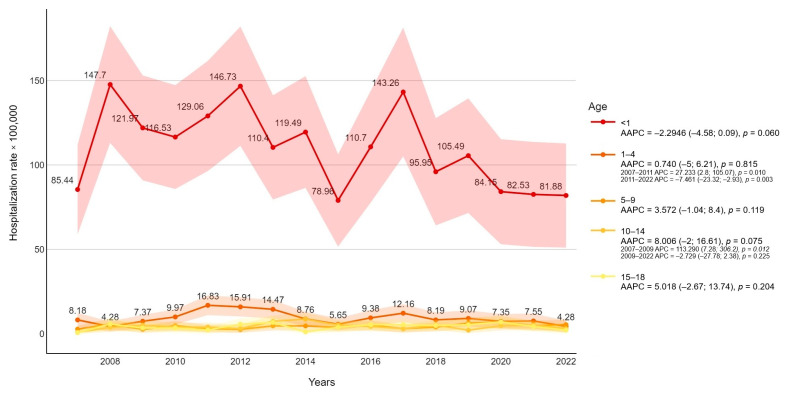
Fungal infection-related hospitalization rate (per 100,000 inhabitants) trends were stratified for age with average annual percentage changes (AAPCs) and 95% confidence intervals.

**Figure 2 pathogens-14-00093-f002:**
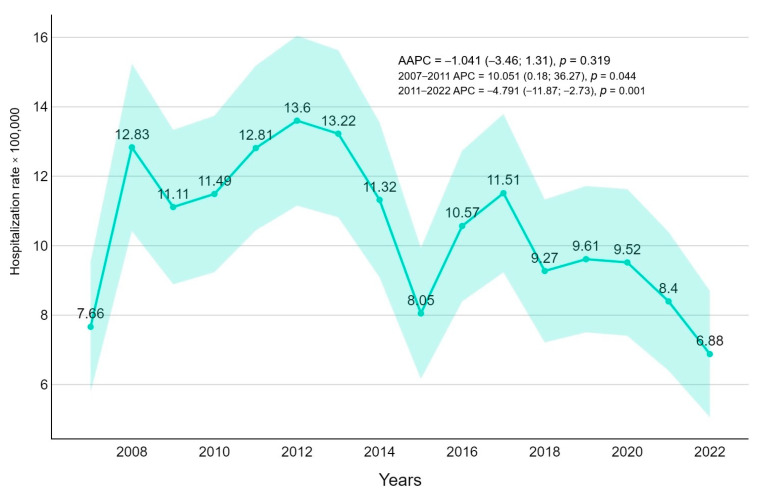
Overall fungal infection-related hospitalization rate (per 100,000 inhabitants) trends with average annual percentage changes (AAPCs) and 95% confidence intervals are presented.

**Figure 3 pathogens-14-00093-f003:**
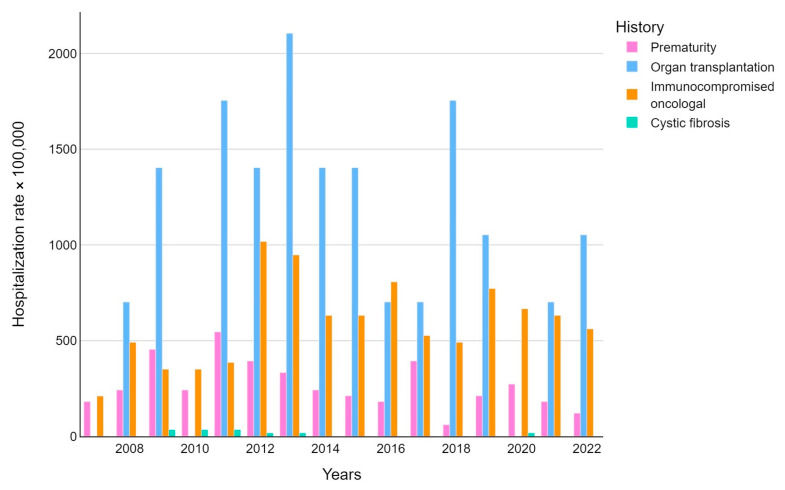
Fungal infection-related hospitalization rate (per 100,000 inhabitants) trends were stratified by disease.

**Figure 4 pathogens-14-00093-f004:**
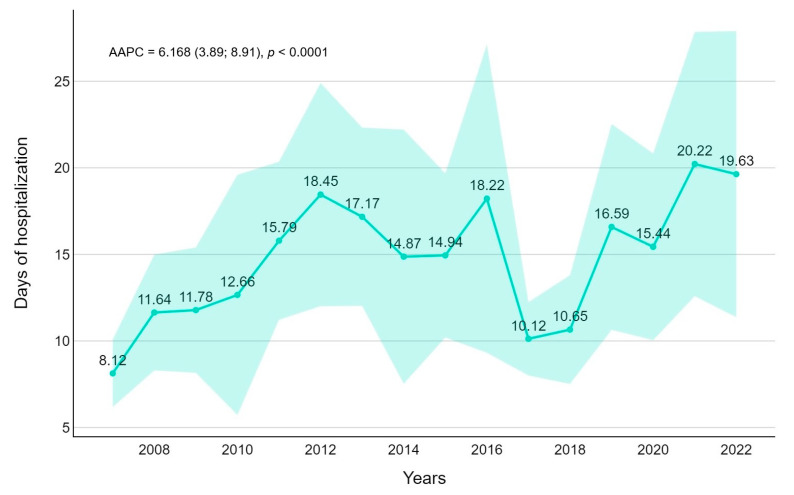
Length of stay trends for fungal infection with average annual percentage changes (AAPCs) and 95% confidence intervals are presented.

**Figure 5 pathogens-14-00093-f005:**
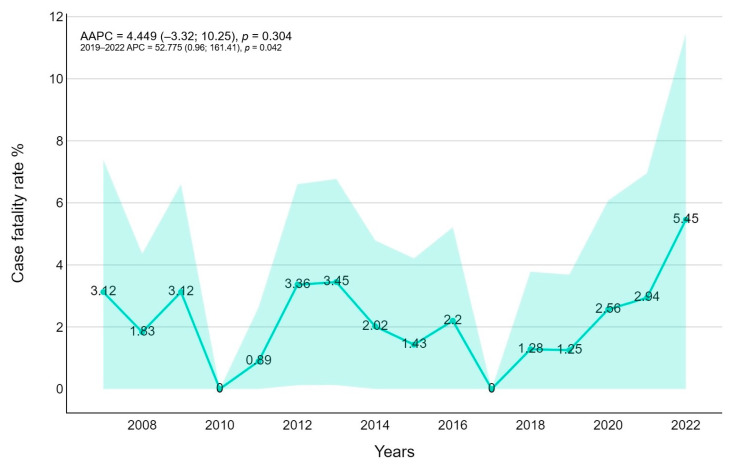
Case fatality trends for fungal infection with average annual percentage changes (AAPCs) and 95% confidence intervals are presented.

**Figure 6 pathogens-14-00093-f006:**
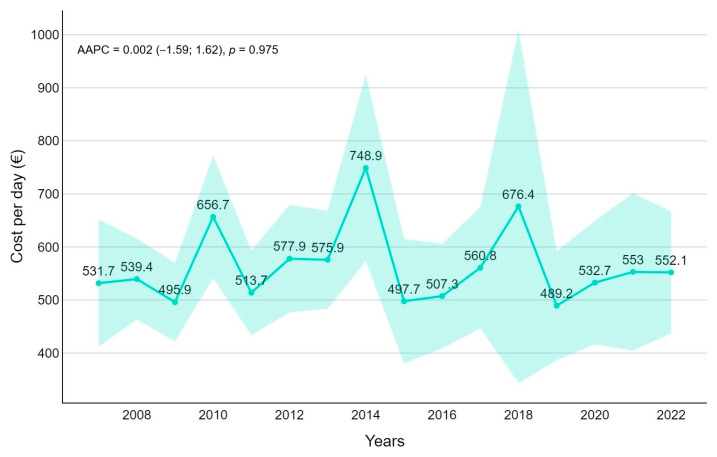
Hospitalization cost trends for fungal infection with average annual percentage changes (AAPCs) and 95% confidence intervals are presented.

**Table 1 pathogens-14-00093-t001:** Fungal infection hospitalization distribution according to age.

Diagnosis (N; %)	<1 Year	1–4 Years	5–9 Years	10–14 Years	15–18 Years	Total
Aspergillosis	4 (0.5)	26 (10)	42 (27)	46 (29)	34 (28)	152 (10.5)
Candidiasis	439 (60)	151 (58)	38 (24)	47 (30)	50 (40.98)	725 (50.5)
Diaper *Candida* infection	58 (8)	15 (6)	1 (0.5)	0 (0)	0 (0)	74 (5)
Dermatomycosis	23 (3)	15 (6)	15 (9.5)	5 (3)	3 (2.5)	61 (4.5)
Neonatal *Candida* infection	175 (24)	/	/	/	/	175 (12)
Mycosis from facultative mycosis	1 (1)	4 (1.5)	0	2 (1)	4 (3)	11 (1)
Mycotic otitis	0	2 (1)	2 (1.5)	(1.26)	(0)	6 (0.5)
Pneumonia due to systemic mycosis	6 (1)	18 (9)	14 (9)	10 (6)	8 (6.5)	56 (4)
Pneumonia due to *Aspergillus*	2 (0.3)	11 (4)	19 (12)	20 (12.5)	5 (4)	57 (4)
*Zygomicosis* infection	0	0	5 (3)	1 (0.5)	1 (1)	7 (0.5)
Other mycoses	24 (3.3)	20 (7.5)	22 (14)	26 (16)	17 (14)	109 (7.6)
Total	732	26 (100)	158 (100)	159 (100)	122 (100)	1433 (100)

**Table 2 pathogens-14-00093-t002:** Fungal infection features and outcome assessments.

Variable	Class	*n*	%	Annual Hospitalization Rate per 100,000	Days of Hospitalization (Mean; Min-Max)	Deaths (n)	CFR %	*p*-Value Hosp. Rate	*p*-Value Days of Hosp.	*p*-Value CFR	*p*-Value Costs
Total	Total	1433	1000	10,839	14.7 (0–368)	30	2094				
Age class	<1	732	51.1	110,552	10.1 (0–320)	6	0.820	<0.0001	<0.0001	0.00165	<0.0001
	1–4	262	18.3	8477	14.5 (0–215)	6	2290				
	5–9	158	11.0	4648	20.1 (0–204)	7	4430				
	10–14	159	11.1	4165	27.8 (0–368)	7	4403				
	15–18	122	8.5	4580	19.1 (0–154)	4	3279				
Sex	F	657	45.8	9536	14.7 (0–220)	16	2435	0.0060	0.3679	0.417	0.5989
	M	776	54.2	11.053	14.8 (0–368)	14	1804				
Year of hospitalization	2007	64	4.5	7662	8.1 (1–50)	2	3.125	<0.0001	0.06545	0.6132	0.2277
	2008	109	7.6	12.832	11.6 (1–100)	2	1835				
	2009	96	6.7	11.112	11.8 (1–119)	3	3125				
	2010	100	7.0	1489	12.7 (0–330)	0	0000				
	2011	112	7.8	12.810	15.8 (0–124)	1	0.893				
	2012	119	8.3	13.600	18.5 (0–302)	4	3361				
	2013	116	8.1	13.224	17.2 (0–220)	4	3.448				
	2014	99	6.9	11,319	14.9 (0–334)	2	2020				
	2015	70	4.9	8049	14.9 (0–111)	1	1429				
	2016	91	6.4	10,567	18.2 (0–368)	2	2198				
	2017	98	6.8	11,513	10.1 (0–48)	0	0000				
	2018	78	5.4	9270	10.7 (0–80)	1	1282				
	2019	80	5.6	9612	16.6 (0–139)	1	1250				
	2020	78	5.4	9519	15.4 (0–154)	2	2564				
	2021	68	4.7	8396	20.2 (0–214)	2	2941				
	2022	55	3.8	6878	19.6 (1.5–204)	3	5455				
Hospitalization during COVID-19 emergency		201	14.0	8396	18.2 (0–214)	7	3.483	<0.0001	0.0021	0.1765	0.3109
Prematurity		141	9.8	242,424	16.6 (0–92)	1	0.709	<0.0001	<0.0001	0.3534	0.6029
Solid organ transplantation		46	3.2	1,403,509	49.8 (0–344)	12	26.087	<0.0001	<0.0001	<0.0001	0.003123
Immunocompromised, onco-hematological		270	18.8	596.491	26.1 (0–368)	16	5.926	<0.0001	<0.0001	<0.0001	<0.0001
Cystic fibrosis		9	0.6	26.315	38.6 (8–119)	0	0.000	0.0137	0.0020	0.9999	0.4003
N° of comorbidities	0	1005	70.1	7621	10.5 (0–330)	10	0.995		<0.0001	<0.0001	<0.0001
	1	392	27.4	2768	23.4 (0–368)	12	3061				
	>1	36	2.5	0.250	39.7 (0–344)	8	22,222				
FI diagnosis	Aspergillosis	152	10.6	0.946	27.5 (0–368)	12	7895		<0.0001	<0.0001	<0.0001
	Candidiasis	725	50.6	5068	10.3 (0–220)	5	0.690				
	Diaper *Candida* infection	74	5.2	0.469	7.7 (0–89)	0	0000				
	Dermatomycosis	61	4.3	0.464	7.2 (1–46)	0	0000				
	Neonatal *Candida* infection	175	12.2	1407	14.0 (1–91)	1	0.571				
	Mycosis from facultative mycosis	11	0.8	0.118	11.1 (0–53)	0	0000				
	Mycotic otitis	6	0.4	0.116	8.7 (2–23)	0	0000				
	Pneumonia due to systemic mycosis	56	3.9	0.240	9.0 (0–65)	0	0000				
	Pneumonia due to *Aspergillus*	57	4.0	0.360	35.5 (0–154)	8	14.035				
	*Zygomicosis* infection	7	0.5	0.182	25.6 (3–57)	0	0.000				
	Other mycoses	109	7.6	0.731	29.0 (0–344)	4	3670				
FI main diagnosis		241	16.8	1687	13.0 (0–145)	3	1245		0.1402	0.4458	0.09509
Emergency admission		931	65.0	7289	12.4 (0–302)	15	1611		<0.0001	0.1227	<0.0001
DRG type	Surgical	77	5.4	0.530	58.0 (0–302)	10	12.987		<0.0001	<0.0001	0.3884
	Medical	1348	94.1	10,393	12.2 (0–368)	19	1409				

CRF: case fatality rate; F: female; M: male; COVID-19: coronavirus infectious disease; FI: fungal infection; DRGs: diagnosis-related groups.

## Data Availability

The data that support the findings of this study are available upon request from the corresponding author.

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
