# Peer review of "Epidemiological Analysis of Fungal Infection Disease in Pediatric Population: Focus on Hospitalization from 2007 to 2022 in Veneto Region in Italy"

_pathogens, 2025, doi:10.3390/pathogens14010093_

Round 1

Reviewer 1 Report

Comments and Suggestions for Authors

Important and well organized and well written study.

1) I would like to see some description on why the general trend is decreasing. Please include your thoughts on the causes for this decreasing trend.

2) Also, some mention of antifungal agents used could be helpful in understanding the decreasing trend and the availabilities of broader spectrum agents during the last years of study as a contributory factor.

3) Among the study limitations please include the limited accuracy and use of ICD codes as another factor.

Author Response

C: Important and well organized and well written study.

R: Thank you very much for the comment

1) I would like to see some description on why the general trend is decreasing. Please include your thoughts on the causes for this decreasing trend.

R: Thank you so much for the comment. We added a possible explanation of FI decreasing in discussion section: “A possible explanation for this reduction trend could be the use of antifungal prophylaxis in specific settings, such as neonatal intensive care units and onco-hematology departments where antifungal prophylaxis protocols are often applied”

2) Also, some mention of antifungal agents used could be helpful in understanding the decreasing trend and the availabilities of broader spectrum agents during the last years of study as a contributory factor.

R: Thank you so much for the comment. We introduced the previous sentence in discussion for the use of antifungal agents. However, the paper is not focused on the prophylaxis and therapy because of a data about antifungal therapy is not applicable. Therefore, we avoided to introduce more deeper this topic.

3) Among the study limitations please include the limited accuracy and use of ICD codes as another factor.

R: Thank you for your comment. We explained better the limits of ICD 9 codes as reported in discussion section.

Reviewer 2 Report

Comments and Suggestions for Authors

Chiusaroli et al., present an analysis of the epidemiological trends of fungal infections in the pediatric population in the Veneto region, Italy.

The work is well designed and presented, the authors mention the main limitation of the study. However, there are some details that can be improved.

Title

The title of the manuscript should be more specific. Since the content focuses solely on fungal diseases in the pediatric population, I suggest including the type of population in the title.

Results

Table 1 mentions that the percentage of other mycoses is 7.5, but in reality it is 7.6 (109/1433)

It is important to at least mention which mycoses were included within “other mycoses”, since the percentage they represent is not negligible.

Figures

The figure legend should be more explanatory. Consider that each figure must be explanatory and understandable.

Discussion

It would be interesting if their results were contrasted with data, if they exist, from Italy rather than the United States.

Lines 29, 30, 46, 49, 54, 209, 218: spp. it is not in italics

Author Response

Chiusaroli et al., present an analysis of the epidemiological trends of fungal infections in the pediatric population in the Veneto region, Italy.
The work is well designed and presented, the authors mention the main limitation of the study. However, there are some details that can be improved.

R: Thank you for your comment.

Title
The title of the manuscript should be more specific. Since the content focuses solely on fungal diseases in the pediatric population, I suggest including the type of population in the title.

R: Thank you for the suggestion. We provided to add a specific information about the pediatric population in the title. The new title is “Epidemiological analysis on fungal infection disease in pediatric population: focus on hospitalization from 2007 to 2022 in Veneto region in Italy”

Results
Table 1 mentions that the percentage of other mycoses is 7.5, but in reality it is 7.6 (109/1433).
It is important to at least mention which mycoses were included within “other mycoses”, since the percentage they represent is not negligible.

R: Thank you for the comment. The percentage are often rounded. We modified the specific percentage. Regarding the second question, the definition “other mycosis” is indicated in hospital discharge records (HDRs) and this is not changeable. We know the analysis of HDR has many limitations that have been indicated in the discussion as the other Reviewer suggested.

Figures
The figure legend should be more explanatory. Consider that each figure must be explanatory and understandable.
R: Thank you for your suggestion. We explained better the comments for each figure.

Discussion
It would be interesting if their results were contrasted with data, if they exist, from Italy rather than the United States.
R: Thank you for the comment; We reported in discussion many studies regarding the similar analysis (HDRs) in United States comparing the general and specifics trends for disease and cost. Unfortunately, the cost analysis can be effectively compared due to the different health care systems between Italy and USA. 

Lines 29, 30, 46, 49, 54, 209, 218: spp. it is not in italics
R: Thank you for the suggestion. We modified in italics.
